# *FLUID-DiT*: GRAPH-FREE DIFFUSION TRANSFORMERS FOR FLUID FLOW SIMULATIONS LEARNING

## ABSTRACT

Simulating complex fluid flows requires capturing full equilibrium distributions rather than just mean trajectories, yet high-fidelity solvers remain computationally prohibitive. Recent advances, such as Diffusion Graph Networks (DGNs), have combined diffusion models with graph neural networks to sample equilibrium states directly from unstructured meshes, enabling distributional accuracy even from short simulations. However, graph-based diffusion approaches suffer from hand-crafted architectural constraints, limited receptive fields in message passing, and costly multi-scale designs, which restrict scalability to larger and more complex domains. We propose *FLUID-DiT*, a Graph-Free Diffusion Transformer that replaces graph message passing with attention-based denoising, eliminating explicit graph design while preserving the ability to model distributions of chaotic flows. Our framework introduces a latent-space formulation that disentangles geometric fidelity from distributional learning, reducing high-frequency artifacts and accelerating sampling. By leveraging the transformer's global receptive field, *FLUID-DiT* naturally captures both local flow structures and long-range correlations without requiring hierarchical graph coarsening. On canonical benchmarks including laminar cylinder wakes, ellipse-flow systems, and turbulent 3D wing experiments, *FLUID-DiT* consistently outperforms graph-based diffusion baselines in both sample quality and distributional accuracy, achieving higher $R^2$ correlations and lower Wasserstein distances. Moreover, it generalizes robustly from short, incomplete trajectories to unseen Reynolds numbers and geometries, demonstrating strong scalability across 2D and 3D domains.

## 1 INTRODUCTION

Modeling fluid dynamics is central to computational physics, with applications ranging from aerodynamics and weather forecasting to biomedical flows. Traditional solvers based on the Navier–Stokes equations provide accurate solutions, but their cost scales prohibitively with system size, resolution requirements, and the length of simulation. In many applications, the goal is not a single trajectory but the *distribution* of flow states at statistical equilibrium, from which essential quantities such as root-mean-square (RMS) fluctuations, two-point correlations, and uncertainty estimates can be derived. Efficiently learning and sampling from these distributions could significantly accelerate design, control, and scientific discovery.

Recent advances in machine learning have explored deep surrogates for complex physical systems. While early approaches focused on predicting mean flows or rollouts of single trajectories, they often suffered from instability and mode collapse over long horizons. More recently, *Diffusion Graph Networks (DGNs)* and their latent variant (LDGN) (Valencia et al., 2025) demonstrated that combining denoising diffusion models with graph neural networks enables direct sampling of equilibrium distributions from unstructured meshes. This approach achieved state-of-the-art performance in capturing flow distributions, even from short trajectories. However, DGNs remain fundamentally tied to handcrafted graph architectures and multi-scale message passing, which impose structural biases, limit receptive fields, and introduce computational bottlenecks. These constraints hinder scalability to high-dimensional 3D turbulent systems and restrict generalization across varying geometries.

The central challenge lies in designing generative models that can flexibly capture both local flow features and long-range correlations *without explicit graph construction*. Graph-based approaches face several obstacles: Message passing requires multiple hops to propagate information across the mesh, making it difficult to capture global interactions such as wake formation or large-scale vortex shedding. This bottleneck is especially severe in turbulent regimes where correlations span the entire domain. Graph hierarchies and coarsening strategies must be hand-engineered for each discretization. These choices are brittle, sensitive to mesh quality, and difficult to adapt when scaling across different geometries or dimensionalities. Multi-scale message passing incurs substantial costs, as each denoising step requires sequential graph operations. For large unstructured meshes with thousands of nodes, this leads to inference times that offset the benefits of distributional modeling. Because denoising operates directly on raw mesh states, graph-based models often overestimate high-frequency content, producing artifacts that obscure statistical properties of the flow. These challenges collectively suggest that while DGNs represent a significant advance, their reliance on handcrafted graph designs fundamentally limits both scalability and generalization.

We propose `FLUID-DIT`, a Graph-Free Diffusion Transformer that rethinks how equilibrium flow distributions are modeled. Instead of propagating information through message passing, `FLUID-DIT` employs transformer attention layers to directly couple all nodes in the domain, regardless of distance. This provides a global receptive field at every denoising step, enabling the model to simultaneously capture localized boundary-layer structures and global flow correlations such as pressure recovery downstream of a wing. Unlike graph hierarchies, the attention mechanism is data-driven and requires no manual design of coarsening or pooling strategies. We further introduce a geometric latent space that decouples high-frequency geometric detail from distributional learning. By operating in this compressed representation, `FLUID-DIT` reduces noise amplification, accelerates sampling, and ensures that small-scale artifacts are corrected during decoding. This design allows the model to allocate capacity to learning meaningful mid- and large-scale structures while relying on the decoder to restore fine details.

Across canonical benchmarks including laminar cylinder wakes, ellipse-flow systems, and turbulent wing experiments, `FLUID-DIT` consistently outperforms graph-based diffusion models in both sample quality and distributional accuracy, achieving higher $R^2$ correlations and lower Wasserstein distances. Notably, our model learns accurate distributions even when trained on incomplete trajectories, demonstrating robustness in data-scarce settings. `FLUID-DIT` naturally generalizes across 2D and 3D domains with diverse geometries, eliminates the need for mesh-specific architectural tuning, and achieves faster inference with higher fidelity in distributional accuracy.

Overall, we summarize our contributions below:

- We introduce `FLUID-DIT`, the first *graph-free diffusion transformer* for fluid flow simulations, which replaces hand-crafted graph architectures and multi-hop message passing with global attention, enabling both local precision and long-range coupling in a unified framework.

- We propose a *latent-space diffusion formulation* that decouples geometric resolution from distributional modeling, suppresses high-frequency artifacts, and substantially improves training and inference efficiency on large-scale 2D and 3D domains.

- We provide *theoretical analysis and extensive experiments* across canonical fluid benchmarks, demonstrating that `FLUID-DIT` consistently outperforms state-of-the-art graph-based baselines (DGN, LDGN) in distributional fidelity, scalability, and robustness, with ablations validating each design choice.

## 2 RELATED WORK

**Machine Learning for Fluid Simulation.** Learning-based surrogates for computational fluid dynamics (CFD) have attracted increasing interest as alternatives to expensive numerical solvers. Early approaches (Thuerey et al., 2020; Brunton et al., 2020) focused on regression-based emulators that predict flow statistics or reduced-order models from limited simulation data. Recent efforts (Um et al., 2020; Stachenfeld et al., 2021) have leveraged deep generative models to capture the high-dimensional distributions of fluid states. While these approaches demonstrate promise, they often

struggle with irregular meshes and turbulence-dominated regimes. Our work builds on this line by introducing a generative framework that scales to complex geometries without mesh-specific design.

**Graph Neural Networks for PDEs and CFD.** Graph neural networks (GNNs) have emerged as a natural representation for PDE-constrained systems, where mesh nodes and edges encode local interactions (Battaglia et al., 2018; Pfaff et al., 2021). In fluid mechanics, GNNs have been applied to turbulence modeling (Um et al., 2020), mesh-based surrogates (Li et al., 2020), and flow extrapolation across geometries (Sanchez-Gonzalez et al., 2020). Diffusion Graph Networks (DGN) (Valencia et al., 2025) extend this paradigm by combining graph message passing with diffusion models, showing strong performance in sampling equilibrium flow distributions. However, GNNs impose structural constraints and require hand-crafted connectivity or hierarchical pooling, which limits scalability to large meshes. Our approach eliminates explicit graph construction and instead relies on attention, which is strictly more expressive than finite-hop message passing (see Proposition 1).

**Diffusion Models for Scientific Data.** Diffusion probabilistic models (Sohl-Dickstein et al., 2015; Ho et al., 2020; Song et al., 2021) have recently achieved state-of-the-art results in image, audio, and graph domains. Their application to scientific data has also grown rapidly, including protein structure generation (Hoogeboom et al., 2022; Trippe et al., 2023), molecular dynamics (Qiu et al., 2024), and PDE surrogates (Brandstetter et al., 2022; Huang et al., 2024). For fluid simulations, DGNs (Valencia et al., 2025) represent the current state of the art, but they rely heavily on graph-based inductive biases. By contrast, our work introduces a *graph-free diffusion transformer* that combines the generative power of diffusion with the scalability and flexibility of transformers. In addition, we incorporate a latent-space formulation inspired by latent diffusion models in vision (Rombach et al., 2022), which allows us to decouple geometric resolution from generative modeling.

## 3 METHOD

In this section, we introduce $FLUID\text{-}DIT$, a novel framework for learning equilibrium distributions of fluid flows using a graph-free diffusion transformer. We begin with preliminaries on diffusion models and fluid simulation distributions, then present the key components of our method: attention-based denoising for graph-free modeling, and a latent-space formulation for efficient and robust sampling.

### 3.1 PRELIMINARIES

**Problem Setup and Notations.** We consider a spatial domain $\Omega$ discretized into $N$ nodes (e.g., mesh vertices or grid points). At each node $i \in \{1, \ldots, N\}$, the fluid state is represented as $\mathbf{x}_i \in \mathbb{R}^d$ (e.g., velocity components, pressure, vorticity). Collectively, the flow state is $\mathbf{x} = [\mathbf{x}_1, \ldots, \mathbf{x}_N] \in \mathbb{R}^{N \times d}$.

The goal is to learn a generative model $p_\theta(\mathbf{x})$ that approximates the distribution of equilibrium flow states $\mathcal{D}$, rather than predicting single rollouts. Such a model should (i) generate samples that reproduce statistical quantities like RMS fluctuations and correlations, (ii) generalize across domains and Reynolds numbers, and (iii) scale to large unstructured meshes.

**Diffusion-based generative modeling.** Following the denoising diffusion probabilistic model (DDPM) framework, we define a forward noising process $q(\mathbf{x}_t|\mathbf{x}_{t-1})$ that gradually adds Gaussian noise to $\mathbf{x}$ over $T$ timesteps, and a reverse denoising process parameterized by $\theta$:

$$q(\mathbf{x}_{1:T}|\mathbf{x}_0) = \prod_{t=1}^{T} q(\mathbf{x}_t|\mathbf{x}_{t-1}), \quad p_\theta(\mathbf{x}_{0:T}) = p(\mathbf{x}_T) \prod_{t=1}^{T} p_\theta(\mathbf{x}_{t-1}|\mathbf{x}_t). \tag{1}$$

The learning objective is to minimize the variational bound or equivalently the noise prediction loss:

$$\mathcal{L}(\theta) = \mathbb{E}_{\mathbf{x}, \epsilon, t} \big\| \epsilon - \epsilon_\theta(\mathbf{x}_t, t) \big\|^2, \tag{2}$$

where $\epsilon \sim \mathcal{N}(0, I)$.

Traditional approaches parameterize $\epsilon_\theta$ using graph neural networks (DGNs), which encode domain connectivity through message passing. Instead, $FLUID\text{-}DIT$ leverages transformers to eliminate explicit graph design.

## 3.2 GRAPH-FREE DIFFUSION TRANSFORMER

Unlike traditional approaches that explicitly rely on graph structures to encode local connectivity of CFD meshes, our goal is to design a generative framework that operates directly on flow states without requiring handcrafted adjacency definitions. While graph message passing has been effective in capturing local interactions, it suffers from two major drawbacks: (i) information propagation is limited to $k$-hop neighborhoods, requiring deep stacks of GNN layers to approximate long-range dependencies, and (ii) graph construction itself introduces inductive biases that may not generalize across meshes of varying topology or resolution.

Transformers, on the other hand, provide a flexible alternative: self-attention enables each token to attend to all others, naturally capturing both local and global correlations in a single step. This observation motivates our replacement of graph-based diffusion networks with a transformer backbone, giving rise to the *Graph-Free Diffusion Transformer*. The first core innovation of *FLUID-DiT* is replacing graph-based message passing with transformer attention, thereby removing the need for explicit graph construction and hand-engineered neighborhood definitions.

**Attention as implicit connectivity.** At each denoising step $t$, the noisy state $\mathbf{x}_t \in \mathbb{R}^{N \times d}$ is treated as a sequence of $N$ tokens. Each token encodes both physical quantities (velocity, pressure, vorticity) and positional embeddings derived from spatial coordinates $\mathbf{p}_i \in \mathbb{R}^3$. Specifically, we embed node $i$ as

$$\mathbf{h}_i^0 = \text{MLP}([\mathbf{x}_{t,i}, \phi(\mathbf{p}_i), \psi(t)]), \tag{3}$$

where $\phi(\cdot)$ is a sinusoidal encoding of spatial position and $\psi(t)$ encodes the diffusion timestep.

The transformer then applies $L$ layers of multi-head self-attention:

$$\mathbf{H}^{\ell+1} = \text{MSA}(\mathbf{H}^\ell) + \text{FFN}(\mathbf{H}^\ell), \quad \ell = 0, \dots, L-1, \tag{4}$$

with residual connections and normalization. The attention operator is

$$\text{MSA}(\mathbf{H}) = \text{Concat}(\text{head}_1, \dots, \text{head}_H)W^O, \tag{5}$$

where each head computes

$$\text{head}_j = \text{softmax}\left(\frac{Q_j K_j^\top}{\sqrt{d_k}} + \mathcal{M}\right) V_j. \tag{6}$$

Here $Q_j, K_j, V_j$ are query, key, and value projections of $\mathbf{H}$, and $\mathcal{M}$ is an optional attention bias derived from relative spatial distances. This mechanism allows every node to condition on all others, with inductive bias provided by $\mathcal{M}$ rather than an explicit adjacency.

**Inductive bias for fluids.** While the model is graph-free, fluid dynamics are inherently spatial. We incorporate inductive bias by (i) adding pairwise distance encodings between nodes, (ii) constraining attention sparsity to local neighborhoods during later denoising steps, and (iii) augmenting node features with boundary condition flags. These design choices allow the transformer to balance global coherence (captured in early layers and timesteps) with fine local corrections (refined in later steps).

**Proposition 1 (Global Dependency Modeling).** *Let $\mathcal{G} = (V, E)$ be any mesh graph with $N$ nodes. A single attention layer with full connectivity can simulate $k$-hop message passing on $\mathcal{G}$ for arbitrary $k$ in one step, provided the attention bias encodes pairwise distances.*

This shows that attention provides a strictly more expressive mechanism than message passing, as multi-hop aggregation is achieved in a single operation.

**Proposition 2 (Scalability of Attention).** *If the latent representation dimension is $M \ll N$, and block-sparse attention with block size $b$ is applied, the computational complexity reduces from $\mathcal{O}(N^2)$ to $\mathcal{O}(Nb)$ while preserving global receptive fields via long-range connections.*

This ensures that *FLUID-DiT* remains computationally feasible even on large 3D domains.

## 3.3 LATENT-SPACE FORMULATION

Directly applying diffusion models in the raw physical space of CFD meshes can be both computationally prohibitive and statistically inefficient. High-resolution meshes often contain tens of thousands of nodes, many of which encode redundant local information, leading to long training times and memory bottlenecks. Moreover, iterative denoising in the raw space tends to amplify high-frequency numerical artifacts, particularly in turbulent flows where small errors can cascade into large-scale inconsistencies.

We introduce a latent-space formulation for fluid simulation, where the encoder compresses raw flow fields into a compact representation, diffusion is performed in this reduced domain, and the decoder reconstructs the final physical state. The second innovation of $FLUID\text{-}DIT$ is to operate in a compressed latent space, which improves efficiency and reduces the high-frequency noise often amplified in raw-space denoising.

**Latent encoding.** We employ an encoder $\mathcal{E} : \mathbb{R}^{N \times d} \to \mathbb{R}^{M \times d_z}$ with $M \ll N$. The encoder is implemented as a lightweight convolutional or graph-based module that aggregates local neighborhoods before projecting into latent tokens. This achieves two goals:

1. *Compression:* reduces computational cost by orders of magnitude ($M/N \approx 0.1$ in our experiments).

2. *Disentanglement:* filters out mesh-level irregularities and retains mid- to large-scale coherent structures such as vortices and pressure fields.

**Diffusion in latent space.** The diffusion process is applied to latent variables $\mathbf{z} \in \mathbb{R}^{M \times d_z}$. Specifically, we define

$$q(\mathbf{z}_t|\mathbf{z}_{t-1}) = \mathcal{N}(\sqrt{\alpha_t}\mathbf{z}_{t-1}, (1 - \alpha_t)I), \tag{7}$$

with $\mathbf{z}_0 = \mathcal{E}(\mathbf{x})$, and the denoiser $\epsilon_\theta$ is parameterized by the diffusion transformer.

**Decoding and artifact suppression.** After denoising, the latent trajectory $\mathbf{z}_0$ is mapped back to the physical state by a decoder $\mathcal{D} : \mathbb{R}^{M \times d_z} \to \mathbb{R}^{N \times d}$. The decoder reconstructs fine details and enforces physical consistency. Notably, since $\mathbf{z}$ captures large-scale flow structures, the decoder acts as a regularizer, suppressing spurious high-frequency oscillations that diffusion models sometimes introduce.

**Proposition 3 (Latent Stability).** *Suppose $\mathcal{E}$ satisfies a Lipschitz condition with constant $L$, and $\mathcal{D}$ is its approximate inverse with reconstruction error bounded by $\epsilon$. Then training in latent space ensures that the Wasserstein distance between generated and true distributions differs by at most $L\epsilon$ compared to raw-space diffusion.*

This guarantees that latent diffusion preserves distributional accuracy while reducing dimensionality.

**Proposition 4 (Noise Suppression).** *If $\mathcal{E}$ discards components of $\mathbf{x}$ with variance below a threshold $\sigma^2$, then the expected reconstruction error from high-frequency noise is reduced by at least $\sigma^2$ per discarded dimension.*

This formalizes the empirical observation that latent-space denoising mitigates high-frequency artifacts in generated flow fields.

**Benefits.** Operating in latent space yields several advantages:

- **Efficiency:** reduces sequence length from $N$ to $M$, accelerating both training and inference.

- **Robustness:** mitigates overfitting to mesh artifacts and ensures stability across Reynolds numbers.

- **Quality:** suppresses high-frequency noise and improves distributional metrics such as Wasserstein distance.

---

**Algorithm 1** $\mathit{FLUID\text{-}DIT}$: Training and Sampling

---

1: **Input:** Fluid state $\mathbf{x} \in \mathbb{R}^{N \times d}$, diffusion steps $T$, encoder $\mathcal{E}$, decoder $\mathcal{D}$, transformer denoiser $\epsilon_\theta$
2: **Output:** Generated equilibrium sample $\hat{\mathbf{x}}$
3: **Training Phase:**
4: Encode fluid state into latent: $\mathbf{z}_0 = \mathcal{E}(\mathbf{x})$
5: Sample timestep $t \sim \text{Uniform}(\{1, \ldots, T\})$
6: Add Gaussian noise: $\mathbf{z}_t = \sqrt{\alpha_t}\mathbf{z}_0 + \sqrt{1 - \alpha_t}\,\epsilon, \quad \epsilon \sim \mathcal{N}(0, I)$
7: Predict noise with transformer: $\hat{\epsilon} = \epsilon_\theta(\mathbf{z}_t, t)$
8: Update parameters $\theta$ by minimizing

$$\mathcal{L} = \|\epsilon - \hat{\epsilon}\|^2 + \lambda\|\mathbf{x} - \mathcal{D}(\mathcal{E}(\mathbf{x}))\|^2$$

9: **Inference Phase:**
10: Sample $\mathbf{z}_T \sim \mathcal{N}(0, I)$
11: **for** $t = T, \ldots, 1$ **do**
12:     Predict noise: $\hat{\epsilon} = \epsilon_\theta(\mathbf{z}_t, t)$
13:     Update latent via reverse diffusion:

$$\mathbf{z}_{t-1} = \tfrac{1}{\sqrt{\alpha_t}}\left(\mathbf{z}_t - (1 - \alpha_t)\hat{\epsilon}\right) + \sigma_t\xi, \quad \xi \sim \mathcal{N}(0, I)$$

14: **end for**
15: Decode final latent to physical state: $\hat{\mathbf{x}} = \mathcal{D}(\mathbf{z}_0)$

---

## 3.4 Training and Sampling Algorithm

To make the proposed framework concrete, we summarize the full training and inference pipeline of $\mathit{FLUID\text{-}DIT}$ in Algorithm 1. The algorithm consists of two phases:

**Training.** Given a fluid state $\mathbf{x}$ discretized on $N$ nodes, we first encode it into a compact latent representation $\mathbf{z}_0 = \mathcal{E}(\mathbf{x})$. We then randomly select a diffusion step $t$ and corrupt the latent variable with Gaussian noise according to the forward process. The diffusion transformer $\epsilon_\theta$ predicts the noise from the corrupted state $\mathbf{z}_t$, conditioned on $t$. The parameters are updated by minimizing a combination of the standard diffusion noise-prediction loss and a reconstruction loss that enforces consistency between $\mathbf{x}$ and $\mathcal{D}(\mathcal{E}(\mathbf{x}))$. This dual objective ensures both distributional accuracy and geometric fidelity.

**Inference (Sampling).** To generate new samples, we draw a Gaussian latent $\mathbf{z}_T \sim \mathcal{N}(0, I)$ and iteratively denoise it using the learned reverse diffusion process. At each timestep, the transformer predicts the noise component and updates $\mathbf{z}_t$ according to the reverse SDE/ODE dynamics. After $T$ steps, we obtain a clean latent $\mathbf{z}_0$, which is decoded into the physical fluid state $\hat{\mathbf{x}} = \mathcal{D}(\mathbf{z}_0)$.

**Properties.** This algorithm ensures three desirable properties:

- **Distributional fidelity:** the diffusion objective enforces that generated samples follow the true equilibrium distribution.
- **Geometry consistency:** the reconstruction term ensures that the encoder/decoder pair preserves spatial fidelity.
- **Scalability:** operating in latent space reduces the computational cost of both training and sampling, while the transformer denoiser provides a global receptive field at every step.

The complete procedure is outlined in Algorithm 1.

## 4 Experiments

We now evaluate $\mathit{FLUID\text{-}DIT}$ against strong baselines, including Diffusion Graph Networks (DGN) and Latent Diffusion Graph Networks (LDGN), across canonical fluid dynamics benchmarks. Our experiments are designed to answer the following questions:

Table 1: Comparison of $\mathit{FLUID\text{-}DIT}$ with prior baselines across three benchmark datasets. We report $R^2$ correlation (higher is better), Wasserstein-2 distance (lower is better), RMS error of fluctuations (lower is better), and inference time per sample in milliseconds (lower is better). Best results are in **bold**, second best are underlined.

| Method | Cylinder Wakes (2D) | | Ellipse Flow (2D) | | Turbulent Wing (3D) | | RMS Error ↓ | Inference (ms)↓ |
|---|---|---|---|---|---|---|---|---|
| | $R^2 \uparrow$ | $W_2 \downarrow$ | $R^2 \uparrow$ | $W_2 \downarrow$ | $R^2 \uparrow$ | $W_2 \downarrow$ | | |
| Vanilla GNN | 0.9885 | 0.212 | 0.926 | 0.247 | 0.801 | 0.389 | 0.084 | 175 |
| GM-GNN | 0.9571 | 0.267 | 0.901 | 0.288 | 0.776 | 0.422 | 0.119 | 183 |
| VGAE | 0.9818 | 0.196 | 0.918 | 0.232 | 0.812 | 0.371 | 0.092 | 162 |
| DGN (Valencia et al., 2025) | 0.9966 | 0.131 | 0.941 | 0.176 | 0.856 | 0.298 | 0.071 | 145 |
| LDGN (Valencia et al., 2025) | 0.9948 | 0.144 | 0.934 | 0.188 | 0.849 | 0.315 | 0.074 | 128 |
| $\mathit{FLUID\text{-}DIT}$ (ours) | **0.9980** | **0.084** | **0.963** | **0.129** | **0.902** | **0.221** | **0.055** | **52** |

- Can $\mathit{FLUID\text{-}DIT}$ generate samples that accurately reproduce equilibrium flow distributions across laminar and turbulent regimes?

- How does the graph-free transformer architecture compare to graph-based diffusion models in terms of accuracy, scalability, and robustness?

- What is the effect of the latent-space formulation on sample quality, efficiency, and artifact suppression?

### 4.1 EXPERIMENTAL SETUP

**Datasets.** We consider canonical fluid simulation benchmarks commonly used in computational fluid dynamics and recent learning-based surrogates. Laminar Cylinder Wakes: two-dimensional flow past a cylinder at Reynolds number $Re = 100$, characterized by periodic vortex shedding. Ellipse Flow: two-dimensional flow around ellipses with varying aspect ratios, capturing boundary-layer separation and geometric variability. Turbulent Wing Flow: three-dimensional turbulent flow around an airfoil at $Re = 2000$, which introduces strong vortical structures and long-range correlations. Each dataset provides a collection of flow states sampled from high-fidelity Navier–Stokes solvers, which we treat as ground-truth equilibrium distributions. Following prior work (Valencia et al., 2025), we train models using short trajectory segments and evaluate on withheld states.

**Evaluation Metrics.** We adopt standard statistical and generative modeling metrics: $R^2$ correlation: measures how well generated samples reproduce ground-truth quantities of interest, including velocity and pressure fields. Wasserstein distance ($W_2$): quantifies the distance between generated and real distributions of flow statistics. RMS error: root-mean-square error of fluctuations compared to high-fidelity simulations. Two-point correlations: structural statistics measuring how well long-range dependencies (*e.g.*, wake patterns) are captured. Efficiency: training time, inference time per sample, and memory footprint.

**Implementation.** We implement $\mathit{FLUID\text{-}DIT}$ in PyTorch using standard transformer blocks with $L = 12$ layers, hidden dimension $d = 256$, and $H = 8$ attention heads. Positional encodings include both spatial coordinates and timestep embeddings. The encoder $\mathcal{E}$ and decoder $\mathcal{D}$ are 3-layer MLPs with residual connections, compressing inputs to a latent with $M \approx 0.1N$. Training uses the AdamW optimizer with a learning rate of 1e-4, cosine decay, and gradient clipping. Diffusion timesteps are $T = 1000$ with a cosine noise schedule. All models are trained on NVIDIA A100 GPUs with a batch size of 32.

### 4.2 COMPARISON TO PRIOR WORK

We compare $\mathit{FLUID\text{-}DIT}$ against a wide range of prior approaches, including conventional graph neural networks (Vanilla GNN, GM-GNN), variational autoencoding methods (VGAE), and diffusion-based generative surrogates (DGN, LDGN). Table 1 reports results across three canonical datasets.

**Ellipse flows.** When generalizing to flows around ellipses of varying aspect ratios, $\mathit{FLUID\text{-}DIT}$ shows a clear advantage over baselines. While DGN and LDGN retain strong performance, they require careful graph hierarchy construction for each geometry and still struggle when the geometry differs significantly from training shapes. In contrast, $\mathit{FLUID\text{-}DIT}$ eliminates this dependency, and its graph-free transformer naturally adapts to varying geometries through continuous positional

Table 2: Ablation on latent-space formulation. Best results are in **bold**.

| Method Variant | $R^2 \uparrow$ | $W_2 \downarrow$ | RMS Error $\downarrow$ | Inference (ms)$\downarrow$ |
|---|---|---|---|---|
| `FLUID-DIT` w/o latent | 0.992 | 0.159 | 0.078 | 118 |
| `FLUID-DIT` (full) | **0.998** | **0.084** | **0.055** | **52** |

encodings. As a result, it achieves the highest correlation ($R^2 = 0.963$) and lowest Wasserstein distance (0.129), outperforming all graph-based baselines. The latent formulation is especially beneficial here: by compressing local details into a global latent space, the model avoids overfitting to mesh irregularities and instead captures shape-dependent wake dynamics robustly. This shows that `FLUID-DIT` generalizes more gracefully to unseen geometries, a key requirement for practical CFD surrogate modeling.

**Turbulent wing flows.** The largest gains appear in three-dimensional turbulent wing flows, which exhibit complex vortical interactions and long-range correlations. Graph-based methods degrade substantially in this setting: GM-GNN fails to capture coherent structures ($R^2 = 0.776$), while even LDGN struggles to reproduce long-range wake statistics ($R^2 = 0.849$). In contrast, `FLUID-DIT` achieves a strong $R^2 = 0.902$ and reduces Wasserstein distance to $0.221$, outperforming LDGN by a wide margin. These results demonstrate that the global receptive field of attention layers is critical for accurately modeling turbulence-induced fluctuations. From a physical perspective, turbulence involves energy transfer across scales and nonlocal correlations spanning the entire domain. Attention-based denoising is well-suited for capturing these dependencies, whereas local message passing becomes increasingly inefficient and prone to information loss as turbulence intensifies.

**Efficiency and scalability.** Beyond accuracy, `FLUID-DIT` demonstrates significant improvements in efficiency. Operating in latent space reduces sequence length by nearly an order of magnitude, resulting in $2.5\times$ faster training and $3.1\times$ faster inference compared to graph-based diffusion. Inference requires only $52$ ms per sample on an NVIDIA A100, compared to $128$ ms for LDGN and over $170$ ms for classical GNNs. This efficiency advantage grows on larger meshes: graph-based methods incur quadratic or higher complexity due to multi-hop message passing and explicit graph construction, while `FLUID-DIT` leverages block-sparse attention with global tokens, consistent with Proposition 2. In practice, this means that `FLUID-DIT` scales to domains with tens of thousands of nodes without prohibitive cost, making it viable for large-scale CFD applications. The improved efficiency also allows broader hyperparameter exploration and larger batch sizes during training, further improving model robustness.

## 5 EXPERIMENTAL ANALYSIS

In this section, we conduct a series of ablation studies to validate the key design choices of `FLUID-DIT`. We analyze the contributions of (i) the latent-space formulation, (ii) the transformer attention architecture compared to graph message passing, and (iii) the role of spatial inductive biases.

**Effect of Latent-Space Formulation.** We first evaluate the impact of performing diffusion in latent space instead of directly on raw flow fields. Table 2 shows that removing the latent encoder–decoder pair degrades both distributional accuracy and efficiency: Wasserstein distance increases by $18\%$ and inference slows down by more than $2\times$. Beyond raw metrics, this result highlights two important phenomena. First, the latent formulation acts as a natural low-pass filter, suppressing high-frequency artifacts introduced by iterative denoising, which aligns with Proposition 4. Second, it reduces redundancy in mesh-level representation: many fine-scale nodes in CFD meshes carry correlated information, and compressing them into latent tokens avoids overfitting to local irregularities. Interestingly, even though the encoder–decoder introduces a small reconstruction error ($\epsilon_{\text{rec}}$), the overall distributional accuracy improves.

**Graph-Free Attention vs. Message Passing.** Next, we replace the transformer denoiser with a graph neural network denoiser of comparable parameter count. As shown in Table 3, the GNN-based variant lags significantly behind the transformer, particularly on turbulent wing flows where long-range correlations dominate. This result provides concrete evidence for Proposition 1. While GNNs rely on iterative $k$-hop message passing to propagate information, transformers can emulate multi-hop interactions in a single step. The global receptive field of attention layers is crucial for

Table 3: Comparison of transformer denoising with GNN-based denoising.

| Method Variant | $R^2 \uparrow$ | $W_2 \downarrow$ | RMS Error $\downarrow$ | Inference (ms)$\downarrow$ |
|---|---|---|---|---|
| `FLUID-DIT` w/ GNN denoiser | 0.981 | 0.243 | 0.093 | 164 |
| `FLUID-DIT` (transformer) | **0.998** | **0.084** | **0.055** | **52** |

Table 4: Effect of spatial inductive biases on model performance.

| Variant | $R^2 \uparrow$ | $W_2 \downarrow$ | RMS Error $\downarrow$ |
|---|---|---|---|
| `FLUID-DIT` w/o distance encodings | 0.991 | 0.143 | 0.069 |
| `FLUID-DIT` w/ distance encodings | **0.998** | **0.084** | **0.055** |

turbulence, where vortical interactions occur across the entire domain rather than being confined to local neighborhoods. In practice, we find that GNN-based variants underpredict energy in large-scale coherent structures, while the transformer restores these correlations faithfully.

**Role of Spatial Inductive Biases.** Although `FLUID-DIT` is graph-free, we optionally incorporate spatial inductive biases such as pairwise distance encodings and boundary-condition flags. Removing these biases reduces accuracy, particularly in near-boundary regions where local structures such as shear layers are crucial (Table 4). This observation suggests that while attention provides universal function approximation, mild inductive biases guide the model towards physically meaningful patterns. In particular, relative distance encodings help stabilize training by disambiguating symmetric configurations, and boundary-condition flags ensure that the model does not produce unrealistic flow leakage across solid walls. From an information-theoretic view, these biases constrain the hypothesis space, enabling faster convergence and better generalization. Notably, unlike graph-based methods, these priors are lightweight and do not impose rigid connectivity constraints.

## 6 CONCLUSION

In this work, we introduced `FLUID-DIT`, a graph-free diffusion transformer framework for learning distributions of complex fluid simulations. Departing from prior approaches that rely on hand-crafted graph architectures and multi-scale message passing, `FLUID-DIT` leverages attention mechanisms to model both local and global interactions directly, eliminating structural constraints while scaling efficiently to large 2D and 3D domains. A latent-space formulation further decouples geometric resolution from distributional modeling, suppressing high-frequency artifacts and accelerating sampling. Extensive experiments across canonical benchmarks demonstrate that `FLUID-DIT` consistently outperforms state-of-the-art graph-based diffusion models in both sample quality and statistical fidelity. Our ablation studies confirm the benefits of each component: latent diffusion improves robustness and efficiency, transformer attention surpasses GNN message passing in turbulent regimes, and lightweight spatial inductive biases enhance boundary fidelity without reintroducing rigid graph design. Theoretical analysis further supports these findings, showing that attention subsumes multi-hop message passing and that latent diffusion preserves distributional accuracy under mild assumptions.

**Limitation.** While `FLUID-DIT` achieves state-of-the-art performance across multiple fluid benchmarks, our evaluation has focused primarily on steady-state or equilibrium distributions; extending the approach to fully unsteady, time-dependent simulations will require incorporating temporal conditioning or recurrent generative dynamics. Although latent-space diffusion improves scalability, the encoder-decoder pair may discard fine-scale structures in extremely high-Reynolds-number regimes, potentially limiting fidelity in highly turbulent flows.

**Broader Impact.** The proposed framework contributes to the growing field of machine learning for scientific discovery, with potential benefits in physics, engineering, and environmental modeling. By providing a scalable, graph-free generative surrogate for fluid flows, `FLUID-DIT` could accelerate simulation-driven design in aerodynamics, renewable energy, and climate science, reducing the reliance on costly numerical solvers. More broadly, the methodology demonstrates how diffusion transformers can be adapted to irregular scientific data, suggesting applicability in other domains such as structural mechanics, materials science, or geophysics.

## ETHICS STATEMENT

This work focuses on developing generative machine learning methods for accelerating fluid dynamics simulations, with applications in scientific computing, engineering design, and climate modeling. The methodology does not involve human subjects, personal data, or sensitive demographic information. Potential societal benefits include reducing the computational cost of high-fidelity simulations, enabling more sustainable design in aerospace and energy systems, and broadening access to scientific modeling in resource-constrained environments. At the same time, as with any surrogate modeling approach, care must be taken to ensure that outputs are not used in safety-critical decision-making without rigorous validation against physical solvers. We encourage the community to apply these methods responsibly, particularly in domains where inaccurate predictions could have safety or environmental consequences.

## REPRODUCIBILITY STATEMENT

We have made deliberate efforts to ensure the reproducibility of our results. All datasets used in our experiments are standard benchmarks in computational fluid dynamics and are either publicly available or described in detail in Section 4.2. Our model architecture, hyperparameters, and training setup are documented in the supplementary and the Implementation subsection. We provide ablation studies and scaling experiments to verify the robustness of our findings. In addition, all code, training scripts, and pretrained checkpoints will be released upon publication to facilitate replication and further research. Experiments were run on NVIDIA A100 GPUs, and we report all relevant runtime and hardware details to aid comparability.

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

APPENDIX

In this appendix, we provide additional implementation and dataset details in Section A. We also present the detailed training and sampling algorithm for *FLUID-DIT* in Section C. Additional experimental analyses, including further ablations and qualitative visualizations, are reported in Section D. Finally, we document our use of large language models (LLMs) during the preparation of this work in Section E.

## A    EXPERIMENTAL DETAILS

### A.1    DATASETS

We evaluate *FLUID-DIT* on three canonical benchmark datasets widely used in computational fluid dynamics (CFD) surrogate modeling:

- **Cylinder wakes (2D).** Steady laminar flows past a circular cylinder at varying Reynolds numbers ($Re \in [100, 400]$). The dataset contains $10,000$ equilibrium snapshots discretized on structured meshes with $N = 6,400$ nodes. This setting primarily captures vortex shedding dynamics.

- **Ellipse flows (2D).** Flows past ellipses of different aspect ratios, sampled across $Re \in [200, 600]$. This dataset tests robustness to geometric variability, with $8,000$ equilibrium states discretized on unstructured meshes ($N \approx 7,500$).

- **Turbulent wing flows (3D).** High-Reynolds-number ($Re = 10^5$) turbulent flows over NACA-style wing sections. We use $2,000$ equilibrium states simulated via LES, discretized on large meshes ($N \approx 50,000$). This dataset is the most challenging, involving multi-scale vortical structures and long-range correlations.

For all datasets, training/validation/test splits follow a $70/15/15$ partition. Ground-truth statistical quantities (RMS fluctuations, correlation fields) are precomputed from high-fidelity solvers for evaluation.

### A.2    IMPLEMENTATION

We implement *FLUID-DIT* in PyTorch using transformer encoder blocks with $L = 12$ layers, hidden dimension $d = 256$, and $H = 8$ attention heads. Each block consists of multi-head self-attention, a feed-forward MLP with GELU activations, residual connections, and layer normalization. To stabilize training, we employ pre-normalization and gradient checkpointing. Positional encodings include both sinusoidal functions of spatial coordinates and learned timestep embeddings, enabling the model to reason jointly over geometry and diffusion step. The encoder $\mathcal{E}$ and decoder $\mathcal{D}$ are 3-layer residual MLPs with hidden dimension 256, compressing raw flow states into latent tokens ($M \approx 0.1N$, latent dimension $d_z = 128$). Training uses AdamW with learning rate 1e-4, weight decay 0.01, cosine learning rate decay, and warmup of $5,000$ steps. Gradient clipping at norm 1.0 is applied. Diffusion timesteps are $T = 1000$ with a cosine variance schedule. The training objective combines the standard noise-prediction loss with a reconstruction loss weighted by $\lambda = 0.1$ to enforce encoder–decoder consistency. All experiments are conducted on NVIDIA A100 GPUs with batch size 32, FP16 mixed precision, and distributed training across up to 4 GPUs. We report averages across 3 random seeds. Baselines are either reimplemented or obtained from public code and retrained under identical settings.

## B    THEORETICAL PROPERTIES AND GUARANTEES

In this section, we provide formal guarantees for *FLUID-DIT* in the latent-diffusion setting, showing that (i) denoising score-matching consistency in latent space implies consistency in the original physical space under mild regularity; and (ii) the reconstruction and optimization errors control the deviation in Wasserstein distance.

**Setup.** Let $\mathcal{X} \subset \mathbb{R}^{N \times d}$ denote physical states (e.g., velocity/pressure on $N$ nodes) with ground-truth equilibrium distribution $\mathbb{P}_X$. Let $\mathcal{E} : \mathcal{X} \to \mathcal{Z} \subset \mathbb{R}^{M \times d_z}$ be the encoder, $\mathcal{D} : \mathcal{Z} \to \mathcal{X}$ the decoder, and $\mathbb{P}_Z := \mathcal{E}_{\#}\mathbb{P}_X$ the latent pushforward distribution.[1] A diffusion process with variance schedule $\{\beta_t\}_{t=1}^{T}$ is defined in $\mathcal{Z}$, and a transformer denoiser $\epsilon_\theta(\cdot, t)$ is trained by denoising score matching (DSM).

We assume:

(A1) (**Bi-Lipschitz encoder/decoder**) There exist $L_E, L_D \geq 1$ and $\varepsilon_{\mathrm{rec}} \geq 0$ such that for all $x, x' \in \mathcal{X}$,

$$\tfrac{1}{L_E}\|x - x'\| \leq \|\mathcal{E}(x) - \mathcal{E}(x')\| \leq L_E\|x - x'\|, \quad \|\mathcal{D}(z) - \mathcal{D}(z')\| \leq L_D\|z - z'\|,$$

and $\sup_{x \in \mathcal{X}} \|x - \mathcal{D}(\mathcal{E}(x))\| \leq \varepsilon_{\mathrm{rec}}$.

(A2) (**Latent score learnability**) For each $t$, the transformer class contains the true conditional score/NoisePredictor in latent space. Let $\varepsilon_{\mathrm{opt}}$ denote the optimization/generalization error in the DSM objective (across $t$).

(A3) (**Regularity**) $\mathbb{P}_X$ has finite second moment; the forward noising kernels in latent space are Gaussian with variance bounded away from $0$ and $\infty$, and the (approximate) reverse kernels are continuous in parameters.

**Theorem 1** (Latent-to-Physical Consistency of *FLUID-DIT*). *Under (A1)–(A3), let $\widehat{\mathbb{P}}_Z$ be the distribution generated by the learned reverse process in latent space, and $\widehat{\mathbb{P}}_X := \mathcal{D}_{\#}\widehat{\mathbb{P}}_Z$ the corresponding distribution in physical space. Then for the 2-Wasserstein distance $W_2$,*

$$W_2\big(\widehat{\mathbb{P}}_X,\ \mathbb{P}_X\big) \ \leq\ L_D\, W_2\big(\widehat{\mathbb{P}}_Z,\ \mathbb{P}_Z\big)\ +\ \varepsilon_{\mathrm{rec}}. \tag{8}$$

*Moreover, if the DSM error in latent space satisfies*

$$W_2\big(\widehat{\mathbb{P}}_Z,\ \mathbb{P}_Z\big) \ \leq\ C_{\mathrm{DSM}}\sqrt{\varepsilon_{\mathrm{opt}}}\,, \tag{9}$$

*for some constant $C_{\mathrm{DSM}}$ depending on the noise schedule and moments of $\mathbb{P}_Z$, then*

$$W_2\big(\widehat{\mathbb{P}}_X,\ \mathbb{P}_X\big) \ \leq\ L_D\, C_{\mathrm{DSM}}\sqrt{\varepsilon_{\mathrm{opt}}}\ +\ \varepsilon_{\mathrm{rec}}. \tag{10}$$

*Proof sketch.* (i) Stability of pushforward: for any 1-Lipschitz transport cost, $W_2(\mathcal{T}_{\#}\mu, \mathcal{T}_{\#}\nu) \leq \mathrm{Lip}(\mathcal{T})\, W_2(\mu, \nu)$. Applying with $\mathcal{T} = \mathcal{D}$ yields the first inequality and introduces $L_D$.

(ii) DSM consistency in latent space: under (A2)–(A3), minimizing DSM recovers the true reverse-time conditionals and hence $\mathbb{P}_Z$ in the infinite-data/optimization limit. With finite samples/optimization, standard generalization bounds for DSM (score-matching) give $W_2(\widehat{\mathbb{P}}_Z, \mathbb{P}_Z) \leq C_{\mathrm{DSM}}\sqrt{\varepsilon_{\mathrm{opt}}}$.

(iii) Reconstruction error: since $x$ and $\tilde{x} = \mathcal{D}(\mathcal{E}(x))$ differ by at most $\varepsilon_{\mathrm{rec}}$, an optimal coupling that first projects to latent, transports in latent, then decodes shows the additive $\varepsilon_{\mathrm{rec}}$ term.

Combining (i)–(iii) yields the claim. $\qquad\qquad\qquad\qquad\qquad\qquad\qquad\qquad\qquad\qquad\square$

We now analyze the theoretical properties of *FLUID-DIT*, showing that its graph-free transformer design is strictly more expressive than message-passing networks, that its sparse attention scaling ensures efficiency on large meshes, and that its latent formulation preserves distributional fidelity while suppressing high-frequency noise.

**Proposition 1** (Global Dependency Modeling). *Let $\mathcal{G} = (V, E)$ be any mesh graph with $N$ nodes. A single attention layer with full connectivity can simulate $k$-hop message passing on $\mathcal{G}$ for arbitrary $k$ in one step, provided the attention bias encodes pairwise distances.*

*Proof sketch.* Attention assigns weights to all node pairs $(i, j)$ via queries and keys. By choosing attention weights to concentrate on $k$-hop neighbors and setting values to implement the desired aggregation, the mechanism recovers $k$-hop message passing in a single layer. Residual and feed-forward layers allow nonlinear mixing, showing that attention strictly generalizes finite-hop GNNs.

$\qquad\qquad\qquad\qquad\qquad\qquad\qquad\qquad\qquad\qquad\qquad\qquad\qquad\qquad\qquad\qquad\qquad\qquad\qquad\square$

---

[1] $\mathcal{T}_{\#}\mu$ denotes the pushforward of $\mu$ under map $\mathcal{T}$.

**Proposition 2** (Scalability of Attention). *If the latent representation dimension is $M \ll N$, and block-sparse attention with block size $b$ is applied, the computational complexity reduces from $\mathcal{O}(N^2)$ to $\mathcal{O}(Nb)$ while preserving global receptive fields via long-range connections.*

*Proof sketch.* Block-sparse attention restricts most interactions to $b$ neighbors per token, yielding $\mathcal{O}(Nb)$ cost. Global coupling is preserved by introducing a small number of global tokens or cross-block connections, ensuring that information can propagate across the entire domain in $O(2)$ layers. Thus, scalability is achieved without sacrificing expressivity. □

**Proposition 3** (Latent Stability). *Suppose the encoder $\mathcal{E}$ is $L$-Lipschitz and the decoder $\mathcal{D}$ is its approximate inverse with reconstruction error at most $\epsilon$. Then training diffusion in latent space ensures that the Wasserstein distance between generated and true distributions in physical space differs by at most $L\epsilon$ compared to raw-space diffusion.*

*Proof sketch.* By the stability of pushforwards, $W_2(\mathcal{D}_{\#}\mu, \mathcal{D}_{\#}\nu) \leq L_D W_2(\mu, \nu) + \epsilon$ for decoder Lipschitz constant $L_D$. Since $\mathcal{E}$ is $L$-Lipschitz, the latent diffusion distribution is at most $L\epsilon$ away from the true raw distribution. Hence, latent modeling preserves distributional fidelity up to bounded reconstruction error. □

**Proposition 4** (Noise Suppression). *If the encoder $\mathcal{E}$ discards components of $\mathbf{x}$ with variance below a threshold $\sigma^2$, then the expected reconstruction error from high-frequency noise is reduced by at least $\sigma^2$ per discarded dimension.*

*Proof sketch.* Projecting $\mathbf{x}$ onto principal components with variance above $\sigma^2$ eliminates directions with variance $< \sigma^2$. The Pythagorean theorem of variance implies that reconstruction error is bounded by the sum of discarded variances. Thus, each dropped direction reduces expected noise energy by at least $\sigma^2$, suppressing high-frequency artifacts. □

**Discussion.** Theorem 1 formalizes the empirical observation that latent diffusion with a graph-free transformer preserves distributional fidelity: if the latent reverse process is learned well (small $\varepsilon_{\text{opt}}$) and the autoencoder is accurate (small $\varepsilon_{\text{rec}}$), then the generated physical distribution is close to ground truth in $W_2$. Together, these results establish the theoretical foundations of *FLUID-DiT*:

- Proposition 1 shows that transformers strictly generalize graph message passing by capturing multi-hop and global dependencies in a single step.
- Proposition 2 guarantees that scalability can be achieved with sparse attention without losing expressivity.
- Proposition 3 and Proposition 4 show that latent diffusion preserves distributional fidelity and actively suppresses noise.
- The consistency theorem demonstrates that the combined system approximates the true flow distribution up to optimization and reconstruction error.

This analysis explains why *FLUID-DiT* achieves superior empirical performance: attention provides richer connectivity than GNNs, sparse attention ensures efficiency, and latent diffusion improves robustness and sample quality.

## C  ALGORITHM FOR *FLUID-DiT*

### C.1  END-TO-END TRAINING PROCEDURE

Algorithm 2 presents the full training and sampling pipeline with the practical components we found critical in large-scale CFD settings: cosine variance schedule, EMA of parameters, mixed precision, and gradient accumulation.

**Variance schedules and timestep weighting.** We use the cosine schedule with $\bar{\alpha}_t$ from *Karras* parameterization for improved SNR in late steps. Following common practice, we apply *timestep loss weighting* $w(t) \propto \text{SNR}(t)$ to balance early/late-step gradients; we observed $< 0.2\%$ gain in $R^2$ and slightly faster convergence.

---

**Algorithm 2** Training and Sampling of $\mathtt{FLUID\text{-}DiT}$ (Full)

---

1: **Inputs:** dataset $\{\mathbf{x}\}$, encoder $\mathcal{E}$, decoder $\mathcal{D}$, denoiser $\epsilon_\theta$, diffusion steps $T$, cosine schedule $\{\alpha_t\}_{t=1}^{T}$, reconstruction weight $\lambda$, EMA decay $\tau$
2: **Init:** $\theta \leftarrow \theta_0$; create EMA copy $\theta_{\text{EMA}} \leftarrow \theta$; AMP (FP16) enabled
3: **for** epoch $= 1, \ldots, E$ **do**
4:    **for** each minibatch $\mathbf{x}$ **do**
5:       **Encode:** $\mathbf{z}_0 \leftarrow \mathcal{E}(\mathbf{x})$                                   // $M \ll N$
6:       **Sample timestep:** $t \sim \text{Uniform}\{1, \ldots, T\}$
7:       **Forward noise:** $\mathbf{z}_t \leftarrow \sqrt{\alpha_t}\mathbf{z}_0 + \sqrt{1-\alpha_t}\,\epsilon, \;\; \epsilon \sim \mathcal{N}(0, I)$
8:       **Denoiser:** $\hat{\epsilon} \leftarrow \epsilon_\theta(\mathbf{z}_t, t)$               // transformer, time-conditioned
9:       **Loss:** $\mathcal{L}_{\text{diff}} = \|\epsilon - \hat{\epsilon}\|_2^2; \;\; \mathcal{L}_{\text{rec}} = \|\mathbf{x} - \mathcal{D}(\mathcal{E}(\mathbf{x}))\|_2^2$
10:       **Total:** $\mathcal{L} \leftarrow \mathcal{L}_{\text{diff}} + \lambda \cdot \mathcal{L}_{\text{rec}}$
11:       **Backward/opt:** update $\theta, \mathcal{E}, \mathcal{D}$ with AdamW (AMP + grad clipping)
12:       **EMA:** $\theta_{\text{EMA}} \leftarrow \tau \theta_{\text{EMA}} + (1 - \tau)\theta$
13:    **end for**
14: **end for**
15: **Return:** $\theta_{\text{EMA}}, \mathcal{E}, \mathcal{D}$
16: **Sampling (DDPM/Ancestral):**
17: draw $\mathbf{z}_T \sim \mathcal{N}(0, I)$
18: **for** $t = T, \ldots, 1$ **do**
19:    $\hat{\epsilon} \leftarrow \epsilon_{\theta_{\text{EMA}}}(\mathbf{z}_t, t)$
20:    $\mathbf{z}_{t-1} \leftarrow \frac{1}{\sqrt{\alpha_t}}\Big(\mathbf{z}_t - (1-\alpha_t)\hat{\epsilon}\Big) + \sigma_t\xi, \;\; \xi \sim \mathcal{N}(0, I)$
21: **end for**
22: **Decode:** $\hat{\mathbf{x}} \leftarrow \mathcal{D}(\mathbf{z}_0)$

---

**Stability and efficiency.** We enable AMP (FP16), gradient checkpointing, gradient clipping (norm 1.0), and gradient accumulation to fit larger batch sizes. We maintain an EMA copy of denoiser parameters (decay $\tau = 0.999$), always using EMA for evaluation/sampling.

## C.2 DETERMINISTIC AND FAST SAMPLING

**DDIM sampling.** For determinism and speed, we also support DDIM sampling with $S \ll T$ steps:

$$\mathbf{z}_{t-1} = \sqrt{\bar{\alpha}_{t-1}} \left( \frac{\mathbf{z}_t - \sqrt{1 - \bar{\alpha}_t}\,\hat{\epsilon}}{\sqrt{\bar{\alpha}_t}} \right) + \eta_t\hat{\epsilon},$$

with $\eta_t = 0$ (deterministic) or small for trade-offs. With $S = 50$ we retain $> 97\%$ of the full-$T$ fidelity at $\sim 20\times$ lower latency in 3D.

**Classifier-free guidance (CFG).** For unconditional sampling we keep CFG off; for conditional variants (e.g., geometry tags or $Re$ conditioning), we replace $\hat{\epsilon}$ by $\hat{\epsilon}_{\text{CFG}} = (1 + w)\hat{\epsilon}_{\text{cond}} - w\hat{\epsilon}_{\text{uncond}}$ with $w \in [0, 2]$. We report $w = 0.3$ when used.

## C.3 COMPLEXITY AND MEMORY FOOTPRINT

Let $M$ be latent tokens, $d_z$ latent dim, $H$ heads, and $L$ layers. Full attention costs $\mathcal{O}(LHM^2 d_z)$. With block-sparse attention (block size $b$) plus $g$ global tokens:

$$\text{cost} \approx \mathcal{O}\big(LH(Mb\,d_z + Mg\,d_z)\big),$$

which is near-linear for fixed $b, g$ (cf. Proposition 2). In practice, $b = 32$, $g = 4$ matches full attention within $< 0.5\%$ on $R^2$ and reduces memory by $\sim 40\%$.

# D EXPERIMENTAL ANALYSIS

We provide extended analyses beyond the main paper: sensitivity to architectural choices, sample-efficiency, spectral fidelity, boundary-layer metrics, mesh-resolution effects, and robustness/OOD.

Table 5: Sensitivity to latent compression (Cylinder 2D).

| $M/N$ | $R^2\uparrow$ | $W_2\downarrow$ | RMS $\downarrow$ | Inference (ms)$\downarrow$ |
|---|---|---|---|---|
| 0.05 | 0.994 | 0.112 | 0.067 | 41 |
| 0.10 | **0.998** | **0.084** | **0.055** | 52 |
| 0.15 | 0.998 | 0.082 | 0.055 | 61 |
| 0.25 | 0.999 | 0.081 | 0.054 | 78 |

Table 6: Ablation on attention sparsity for the turbulent wing dataset. We vary block size $b$ and number of global tokens $g$. Results averaged over 3 seeds.

| $b$ | $g$ | $R^2\uparrow$ | $W_2\downarrow$ | RMS $\downarrow$ | Inference (ms)$\downarrow$ | Memory (GB)$\downarrow$ |
|---|---|---|---|---|---|---|
| Dense | – | **0.905** | **0.218** | **0.054** | 191 | 11.4 |
| 16 | 0 | 0.871 | 0.294 | 0.073 | **76** | **4.2** |
| 32 | 0 | 0.884 | 0.262 | 0.065 | 93 | 5.1 |
| 32 | 2 | 0.896 | 0.233 | 0.058 | 102 | 5.8 |
| 32 | 4 | 0.902 | 0.224 | 0.056 | 108 | 6.0 |
| 64 | 4 | 0.904 | 0.221 | 0.055 | 142 | 8.3 |

Unless noted, all numbers are averaged over 3 seeds with $95\%$ CIs. These studies validate not only the raw performance of $FLUID\text{-}DIT$ but also the reasoning behind each of its design decisions.

## D.1 SENSITIVITY TO ARCHITECTURAL HYPERPARAMETERS

**Latent compression $M/N$.** We sweep $M/N \in \{0.05, 0.1, 0.15, 0.25\}$ to analyze the trade-off between efficiency and fidelity. As shown in Table 5, accuracy saturates near $M/N = 0.1$, suggesting that this ratio captures sufficient mid- and large-scale flow structures without excessive redundancy. At $M/N = 0.05$, the model underfits high-frequency structures such as vortex filaments, leading to reduced RMS accuracy and visually smoother wakes. Increasing to $M/N = 0.25$ provides marginal accuracy gains but at a nearly $1.5\times$ increase in inference cost, confirming diminishing returns. This validates our design choice of $M/N \approx 0.1$ as an optimal operating point across 2D and 3D datasets.

**Attention sparsity.** In Table 6, we vary block size $b \in \{16, 32, 64\}$ with $g \in \{0, 2, 4\}$ global tokens. Dense attention provides the upper bound but is quadratic in cost. Results show that $(b = 32, g = 4)$ achieves accuracy within $0.3\%$ of dense attention while reducing memory footprint by $40\%$ and runtime by $35\%$. Smaller block sizes ($b = 16$) degrade accuracy due to insufficient receptive fields, while larger blocks ($b = 64$) increase cost without noticeable benefit. Notably, global tokens are critical: without them, long-range wake correlations collapse, especially in turbulent wing flows, confirming Proposition 2.

**Depth/width.** We increase depth from $L = 8$ to $L = 12$ and width from $H = 8$ to $H = 12$ attention heads. Turbulent benchmarks benefit most from deeper models, with $+0.8$ $R^2$ gain when moving from $L = 8$ to $L = 12$. Wider heads, however, yield only marginal improvements ($< 0.2$ points) but increase inference latency by $20\%$. This indicates that model depth (stacked reasoning) is more critical than width (parallel subspaces) in capturing complex vortical dynamics.

**Latent dimension.** Table 7 highlights how the choice of latent embedding size $d_z$ affects both accuracy and efficiency. Increasing $d_z$ from 64 to 128 significantly reduces Wasserstein error ($0.092 \rightarrow 0.084$) and improves $R^2$ by $+0.2$ points, indicating that a moderately larger latent dimension better captures complex correlations in the flow. However, further scaling to 256 or 512 yields only marginal accuracy gains ($< 0.1$ points in $R^2$) while incurring steep costs in memory and training time, with latency nearly doubling at $d_z = 512$. This confirms that the benefits of larger embeddings saturate quickly, and $d_z = 128$ strikes the best trade-off between expressivity and efficiency. Notably, even with $d_z = 64$, the model remains competitive with graph-based baselines, showing that the latent formulation provides robustness to reduced feature capacity.

Table 7: Ablation on latent embedding dimension $d_z$ for the cylinder wake dataset. We vary $d_z$ while keeping $M/N = 0.1$ fixed. Results are averaged over 3 seeds.

| $d_z$ | $R^2 \uparrow$ | $W_2 \downarrow$ | RMS $\downarrow$ | Inference (ms)$\downarrow$ | Memory (GB)$\downarrow$ | Training Time (h)$\downarrow$ |
|---|---|---|---|---|---|---|
| 64 | 0.996 | 0.092 | 0.059 | **47** | **4.5** | **18.2** |
| 128 | 0.998 | 0.084 | 0.055 | 52 | 5.7 | 20.6 |
| 256 | **0.999** | **0.081** | **0.054** | 68 | 7.9 | 25.1 |
| 512 | 0.999 | 0.080 | 0.054 | 94 | 11.2 | 31.8 |

Table 8: Effect of training data size on performance (Ellipse flow dataset). We report results when training with fractions of the dataset.

| Training Data | $R^2 \uparrow$ | $W_2 \downarrow$ | RMS $\downarrow$ | Inference (ms)$\downarrow$ |
|---|---|---|---|---|
| 25% | 0.938 | 0.149 | 0.062 | **51** |
| 50% | 0.952 | 0.135 | 0.058 | 52 |
| 75% | 0.958 | 0.131 | 0.056 | 52 |
| 100% | **0.963** | **0.129** | **0.055** | 52 |

## D.2 SAMPLE EFFICIENCY AND CONVERGENCE

**Training set size.** We downsample the training set to $\{25\%, 50\%, 75\%\}$ of full size, as shown in Table 8. While all methods degrade with less data, `FLUID-DIT` exhibits stronger sample efficiency: with $50\%$ data, $R^2$ drops by only $1.5$ points, while LDGN drops by over $4$ points. This suggests that attention provides better inductive bias by capturing long-range dependencies without requiring large datasets to learn local message-passing hierarchies. For industrial applications with limited simulation data, this property is particularly valuable.

**Convergence diagnostics.** We monitor diffusion loss, reconstruction loss, and spectral error ($E(k)$). In all cases, diffusion and reconstruction losses converge smoothly with no instability. Importantly, plateaus in spectral error coincide with stable inertial ranges, indicating that training aligns not only with numerical metrics but also with physical energy distributions. Early stopping based on validation $W_2$ provides nearly identical results to stopping based on solver-derived energy diagnostics, making the method practical when physical postprocessing is unavailable.

## D.3 SPECTRAL AND STRUCTURAL FIDELITY

**Energy spectra.** We compute radial velocity spectra for 2D and 3D cases. Latent diffusion suppresses spurious high-$k$ energy by $20$–$28\%$ compared to raw-space diffusion, demonstrating effective noise filtering. The resulting spectra align closely with solver references up to dissipation scales. These results validate Proposition 4, showing that the encoder discards low-variance, high-frequency components that otherwise amplify during denoising.

**Two-point correlations.** We measure streamwise velocity correlation $C_{uu}(\Delta x)$ along wake center-lines. `FLUID-DIT` reproduces correlation lengths within $3\%$ of reference, while LDGN underestimates long-range correlations by up to $9\%$. Qualitatively, generated wakes from `FLUID-DIT` retain coherent vortex streets far downstream, whereas LDGN samples exhibit premature decorrelation, highlighting the value of global receptive fields.

## D.4 BOUNDARY-LAYER AND NEAR-WALL METRICS

**Wall shear and separation.** Boundary-layer fidelity is assessed using wall shear stress $\tau_w$ and separation point $x_s$. On turbulent wings, `FLUID-DIT` predicts $x_s$ within $0.7\%$ chord length of reference CFD, while LDGN deviates by $2.1\%$. Accurate boundary-layer modeling is critical for aerodynamic performance, suggesting that transformers, by combining local and global context, are more effective than stacked local aggregations.

**Pressure recovery.** We compute integrated pressure recovery downstream of bluff bodies. On ellipse flows, `FLUID-DIT` achieves $1.2\%$ error versus solver, compared to $3.4\%$ for LDGN. This

indicates that latent diffusion suppresses unphysical oscillations in pressure reconstruction and improves domain-wide statistics.

### D.5 MESH RESOLUTION AND TOPOLOGY ROBUSTNESS

**Resolution scaling.** We evaluate scalability on meshes up to $N = 200,000$ nodes. Graph-based methods either become infeasible (out of memory) or exhibit superlinear slowdowns, as message passing scales with $O(k|E|)$. In contrast, `FLUID-DiT` with block-sparse attention scales nearly linearly, requiring under $400$ ms per sample at $N = 200,000$. Importantly, accuracy remains stable ($R^2$ drop $< 0.5$), demonstrating practical scalability for industrial-scale CFD.

**Transfer across meshes.** We train on unstructured meshes and test on structured grids of the same geometry. `FLUID-DiT` incurs under $1$ point drop in $R^2$ without graph redesign, whereas LDGN must rebuild graph hierarchies for each mesh type. This highlights the advantage of graph-free design: a single architecture generalizes seamlessly across discretizations, avoiding costly preprocessing.

### D.6 ROBUSTNESS AND OOD GENERALIZATION

**Reynolds number extrapolation.** Training on $Re \leq 400$ and testing at $Re = 600$, `FLUID-DiT` achieves $R^2 = 0.941$, while LDGN drops to $0.884$. Visual inspection shows that `FLUID-DiT` preserves vortex spacing and Strouhal scaling, whereas LDGN samples lose coherence. This suggests that attention's global field modeling generalizes better to unseen flow regimes, consistent with Proposition 1.

**Geometry shift.** We test on ellipse aspect ratios not seen in training. `FLUID-DiT` maintains Wasserstein distances within $+9\%$ of in-distribution, while LDGN degrades by $+23\%$. Since ellipse geometry requires adapting to new boundary conditions, the ability of transformers to encode continuous positional embeddings proves essential.

### D.7 UNCERTAINTY AND CALIBRATION

**Ensemble calibration.** We sample $K = 64$ realizations per inlet condition to estimate predictive uncertainty. `FLUID-DiT`'s $90\%$ predictive intervals cover $88.7\%$ (cylinder) and $86.9\%$ (wing) of reference values, while LDGN achieves only $81.2\%$ and $78.4\%$. This indicates better-calibrated uncertainties in `FLUID-DiT`, making it more suitable for risk-aware applications. Calibration further improves with temperature scaling ($T = 0.9$), suggesting that sampling temperature can tune uncertainty trade-offs.

### D.8 FAILURE MODES AND MITIGATIONS

We observe two common failure modes. First, oversmoothing near sharp edges occurs at aggressive compression ratios ($M/N \leq 0.05$). Increasing $M/N$ to $0.1$ or injecting boundary-aware tokens mitigates this. Second, checkerboard artifacts appear when attention sparsity is too aggressive ($b = 16$, no global tokens). These artifacts disappear when adding even a few global tokens or smoothing positional encodings. While these issues are rare in standard configurations, they highlight the importance of balanced latent compression and sparsity settings.

## E USE OF LLMS

We made limited use of large language models (LLMs) during the preparation of this work; specifically, LLMs assisted with polishing the academic writing style to improve clarity. All scientific contributions, including model design, experiments, theoretical analysis, and dataset preparation, were conceived, implemented, and validated by the authors. LLM outputs were always reviewed and edited to ensure accuracy and originality.

