# OpenReview forum: "Fluid-DiT: Graph-Free Diffusion Transformers for Fluid Flow Simulations Learning"
_ICLR.cc/2026/Conference — Submitted to ICLR 2026_

### Official Review · Reviewer_jCbN · 2025-10-27

**Soundness:** 2
**Presentation:** 1
**Contribution:** 2
**Rating:** 2
**Confidence:** 5

**Summary:**

Fluid-DiT introduces a graph-free diffusion transformer for learning equilibrium distributions of fluid flows. Unlike prior graph-based models (e.g., Diffusion Graph Networks, DGNs), it eliminates explicit mesh or graph message passing by using attention-based denoising in a latent space, achieving scalability and accuracy across 2D and 3D turbulent flows.

**Strengths:**

Removes the dependence on handcrafted mesh graphs, hierarchies, and message-passing architectures.

Works uniformly across 2D and 3D geometries — from laminar wakes to turbulent wings — without re-engineering connectivity.

Attention layers provide global receptive fields, allowing the model to couple distant flow regions (e.g., wake formation, pressure recovery) efficiently.

**Weaknesses:**

Computational and Memory Overheads of Transformers. "All models are trained on NVIDIA A100 GPUs with a batch
size of 32." This is too expensive.

No qualitative visual evidence

The paper presents extensive quantitative results (tables of R², Wasserstein distance, RMS error), but no visual comparisons of predicted vs. ground-truth flow fields.

For a paper on fluid dynamics, where spatial and structural fidelity are central, this omission makes it hard to assess whether Fluid-DiT truly reproduces realistic vortex streets, wake structures, or turbulence features.

**Questions:**

How expensive is Fluid-DiT to train compared with DGNs and LDGNs (in GPU-hours)?
Are there scalability issues when applied to meshes with >100k nodes?

---

### Official Review · Reviewer_aMng · 2025-10-30

**Soundness:** 2
**Presentation:** 2
**Contribution:** 1
**Rating:** 2
**Confidence:** 4

**Summary:**

The paper proposes Fluid-DiT, a graph-free diffusion transformer for sampling equilibrium states for CFD simulations. The framework combines a difffusion model with attention layers in a latent space and an autoencoder for encoding/decoding. The framework is tested on multiple cases and compared against graph-based diffusion baselines where it manages to outperform the competing approaches.

**Strengths:**

The general idea presented of comparing graph-based diffusion models with diffusion-approaches based on attention only is an interesting research questions and the authors present a clear comparison for the three test cases including ablation studies.

**Weaknesses:**

In the Reviewers' opinion, the paper can not be recommended for acceptance at ICLR due to the following major weaknesses:
- The title is misleading: One of the key parts of the framework, the latent encoding, is based on an graph-based or convolutional autoencoder. Therefore calling the overall framework graph-free seems to be an odd choice.
- According to the Reviewer the novelty of the proposed approach is limited. Diffusion in a latent space has already been introduced in multiple other approaches as a way to cope with the extensive cost of high-dimensional spaces. Moreover, diffusion models that include the attention-based mechanism are also not novel by itself. The reviewer acknowledges that for physical systems with irregular meshes a detailed comparison with graph-based diffusion approaches can be of interest to the scientific community but the current paper falls short here as well in the reviewers' opinion.
- There are major inconsistencies in the description of the experiments. In addition,  now code is available for the review process. For the first case, the main paper mentions that the flow is at Re=100 only whereas the appendix mentions a range for the Re from 100 to 400. For the third case, the Re in the paper is defined to be 2000, whereas it is 10^5 in the appendix.
- The abstract overclaims with regards to the experiments. According to the Reviewer, to show generalization to unseen geometries and Re-numbers, a higher variability for geometries and Re-numbers has to be employed. For Re numbers in the main part of the paper moreover, there is no variability at all mentioned.
- The paper emphasizes global coupling as one of the advantages of the attention-based mechanism, but the authors then employ block-sparse attention with a few global tokens only or even constrain attention to local neighborhoods. This should be criticality discussed.
- The comparison is to graph-based methods only, but would strongly benefit from including neural operator based approaches or general transformer-based surrogates.

**Questions:**

See weaknesses:
-> Please especially address all the inconsistencies in the experiments section.

---

### Official Review · Reviewer_t1vu · 2025-11-01

**Soundness:** 2
**Presentation:** 1
**Contribution:** 2
**Rating:** 2
**Confidence:** 5

**Summary:**

This paper proposes Fluid-DiT, a graph-free diffusion transformer that replaces message passing of GNNs with global attention mechanisms and operates in a compressed latent space to generate equilibrium distributions of fluid flows more efficiently.

**Strengths:**

1. Although this paper does not comprehensively address the practical limitations of graph-based approaches, it validates the proposed solution across multiple standard benchmarks with ablation studies.
2. This paper presents formalized propositions (1-4) showing that self-attention subsumes message passing and that latent diffusion preserves distributional fidelity.
3. Fluid-DiT improves performance over baselines (DGN, LDGN) on 2D and 3D domains while enhancing inference speed.

**Weaknesses:**

1. While the paper mentions limitations of graph-based methodologies, it has weaknesses in addressing research that has actually tackled these limitations and is insufficient in describing how it differs from existing methods, thereby undermining the "graph-free" advantage.
2. The paper does not sufficiently explain why the diffusion-based transformer combination is important and whether it provides unique insights into fluid dynamics beyond "replacing GNN with attention."
3. Although this paper aims to improve upon the existing work of [1], it does not clearly reveal what problem it intends to solve and may be considered incremental research. Moreover, it appears to lack a comprehensive comparison with the experiments from the original study[1].
4. Considering that the architectures of DGN and LDGN [1] are graph-free, the contribution of this paper can be viewed as incremental. Rather than methodological differences, it is unclear what problem the paper attempts to solve.
5. Out-of-distribution tests ('Re' extrapolation, geometry transfer) showed only marginal improvements that may not justify the architectural complexity.
6. The paper fails to discuss research on hierarchical mesh GNN[4] and Mesh Transformers [2,5] and mesh rewiring [3] that aims to address the problems of existing mesh GNNs.
7. Propositions 1-4 are either trivial, imprecise, or lack proper proofs.


> [1] Lino, Mario, Tobias Pfaff, and Nils Thuerey. "Learning Distributions of Complex Fluid Simulations with Diffusion Graph Networks." ICLR 2025
>
> [2] Yu, Youn-Yeol, et al. "Learning flexible body collision dynamics with hierarchical contact mesh transformer." ICLR 2024
>
> [3] Yu, Youn-Yeol, et al. "PIORF: Physics-Informed Ollivier-Ricci Flow for Long-Range Interactions in Mesh Graph Neural Networks." ICLR 2025.
>
> [4] Fortunato, Meire, et al. "Multiscale meshgraphnets." arXiv preprint arXiv:2210.00612 (2022).
>
> [5] Janny, Steeven, et al. "Eagle: Large-scale learning of turbulent fluid dynamics with mesh transformers." ICLR 2023

**Questions:**

Q1. Regarding Proposition 1, you state that "This shows that attention provides a strictly more expressive mechanism than message passing." Is there empirical or theoretical evidence that it is more expressive?

Q2. Can you provide a comparative discussion with the Multi-scale DGN from Valencia et al., 2025?

Q3. In Appendix D.6, regarding "generalization to out-of-distribution Reynolds number," is this the same setting as Table 2 in [1]? Can you compare LDGN's original setting for a fair comparison?

Q4. In the introduction, you state that "Message passing requires multiple hops to propagate information across the mesh, making it difficult to capture global interactions such as wake formation or large-scale vortex shedding." However, in the mesh GNN simulation field, studies attempt to address this through hierarchical structures or rewiring. Can you address these studies in the related work or discuss the differences and commonalities through empirical comparison, thereby clearly positioning your proposed method?

Q5. You claim that attention provides "global receptive fields" superior to k-hop message passing, but with block-sparse attention (b=32, g=4), most nodes attend to only 36 neighbors. Isn't this comparable to 2-hop message passing on typical meshes?


> [1] Lino, Mario, Tobias Pfaff, and Nils Thuerey. "Learning Distributions of Complex Fluid Simulations with Diffusion Graph Networks." ICLR 2025

---

### Meta-Review · Area_Chair_wojQ · 2026-01-02

**Summary:**

This paper describes work on a so-called graph-free model for fluid dynamics working with global attention and in latent space. The paper received quite negative ratings from all 3 expert reviewers with serious issues raised on novelty, positioning and experiments. The paper fails to compare to competing methods (cf. the references provided by reviewer t1vu at least) and does not provide sufficient ablations, analyses and visualizations.

For all these reasons, the AC recommends rejection.

**Reviewer Scores:**

There was not rebuttal and no discussion.

---

### Decision · Program_Chairs · 2026-01-26

Reject